# Seventh ISNS Reference Preparation for Neonatal Screening for Thyroid Stimulating Hormone, Phenylalanine, and 17α-Hydroxyprogesterone in Blood Spots

**DOI:** 10.3390/ijns11010013

**Published:** 2025-02-09

**Authors:** Peter C. J. I. Schielen, Dianne Webster, J. Gerard Loeber, James R. Bonham

**Affiliations:** 1International Society for Neonatal Screening, Reigerskamp 273, 3607HP Maarssen, The Netherlands; gerard.loeber@gmail.com; 2National Newborn Screening Laboratory, LabPlus, Health New Zealand Te Whatu Ora, Auckland 1023, New Zealand; diannew@adhb.govt.nz; 3Liggins Institute, University of Auckland, Auckland 1023, New Zealand; 4Clinical Chemistry, Sheffield Children’s NHSFT, Sheffield S10 2TH, UK; j.bonham@nhs.net

**Keywords:** reference preparation, ISNS-RPNS, TSH, thyroid-stimulating hormone, Phenylalanine, 17α-Hydroxyprogesterone

## Abstract

The International Society for Neonatal Screening (ISNS) has supported the standardization of the measurement of key biochemical markers for the neonatal screening of diseases: thyroid-stimulating hormone (TSH) for congenital hypothyroidism, phenylalanine (PHE) for phenylketonuria, and 17α-hydroxyprogesterone (17OHP) for congenital adrenal hyperplasia. These diseases are commonly a part of neonatal screening panels worldwide. The ISNS provides a series of secondary reference materials to the manufacturers of neonatal screening reagents to assist in the production of calibration materials for kits. This technical note describes the manufacture of the seventh combined dried blood spot reference preparation for neonatal screening (RPNS) for these analytes.

## 1. Introduction

Neonatal screening (NS) for congenital disorders is based on the determination of the analytes of interest in dried blood spot (DBS) samples on blood spot collection paper. Kits intended for use in neonatal screening usually have DBS calibrators. Quality-assurance programs for neonatal screening show a large variability, which is partly due to how manufacturers assign analyte values to the calibrators. The International Society for Neonatal Screening (ISNS) prepared a combined reference preparation for neonatal screening (RPNS) for thyroid-stimulating hormone (TSH), phenylalanine (PHE), and 17α-hydroxyprogesterone (17OHP) in dried blood spots (DBSs).

The first ISNS Reference Preparation for Neonatal Screening for TSH, phenylalanine (PHE), and 17OHP in DBSs (first ISNS-RPNS) was first prepared in 2004 and was made available to the manufacturers of kits for neonatal screening and to the organizers of quality-assurance programs [1]. Since then, six additional preparations have been made (Table 1). The latest materials were prepared in June of 2024 with a 5-year expiry (until December 2029) by the Newborn Screening and Molecular Biology Branch (NSMBB) at the Centers for Disease Control and Prevention (CDC, Atlanta, GA, USA), in collaboration with the ISNS.

## 2. Preparation of Seventh ISNS-RPNS

The detailed procedure for the preparation of the ISNS-RPNS has been described in detail elsewhere [1]. The seventh ISNS-RPNS followed a similar procedure. Briefly, anonymously collected blood from healthy adult donors was purchased from a commercial blood bank. The blood products were tested and found to be hepatitis B surface antigen (HBsAg) negative, hepatitis B virus nucleic acid testing (NAT) negative, HIV 1 and 2 antibody negative, HIV NAT negative, hepatitis C virus antibody negative, hepatitis C virus NAT negative, and syphilis negative. In addition, all human red cell products were negative for Chagas Disease (T.cruzi) and West Nile Virus NAT. Units of blood group-compatible red blood cells were pooled and mixed with the purchased serum of a compatible blood group. The whole-blood matrix was adjusted to 50% hematocrit and divided into five portions. The portions were enriched with (i) TSH (81/565, National Institute for Biological Standards and Control, Potters Bar, UK), (ii) L-phenylalanine (Sigma-Aldrich, St. Louis, MI, Catalog #P2126-100G), and (iii) 17α-hydroxyprogesterone (Sigma Aldrich, Catalog #: H5752-5G). After enrichment, the blood was dispensed onto blood spot paper Grade 903 supplied by Eastern Business Forms (Greenville, SC, USA) using an automated liquid-handling system. Two 75 µL spots for each calibrator (A–E) were spotted on each card. The cards were dried overnight at ambient temperature, packed in zip-closure bags with desiccant, and stored at −20 °C. The concentration of each analyte in the five calibrators was measured to ensure the analyte enrichment was reached. Table 2 gives the enrichment levels for each analyte in the five calibrators.

## 3. Analyte Recovery of the Seventh ISNS-RPNS

The calibrators of the seventh ISNS-RPNS were analyzed by methods performed at NSMBB and the recoveries ranged from 92.9% to 110.6% for TSH, from 90.3% to 96.3% for PHE, and from 92.2% to 109.7% for 17OHP, taking into account the endogenous concentrations of TSH, PHE, and 17OHP in the whole-blood matrix. These recoveries are characteristic of these kinds of preparations. The typical concentrations of the zero-enrichment levels for the whole-blood matrix used for the seventh ISNS-RPNS were 78.8 μmol/L for PHE (derivatized tandem mass spectrometry (tandem-MS) commercial assay) or 69.1 μmol/L (non-derivatized tandem-MS) commercial assay), 0.8 mIU/L for TSH, and 2.6 nmol/L for 17OHP. Significantly more testing of the RPNS materials would need to be carried out to determine whether the differences in recovery are significant and to set firmer values. The seventh ISNS-RPNS, therefore, is not a secondary reference material, and the proper way to use this RPNS is to determine the linear relationship between the measurements of the various enrichment levels (see also [2]).

Therefore, the nominal values of the three components as stated on the labels should be used for calibration purposes.

## 4. Conclusions

The seventh ISNS-RPNS is available, upon request, to manufacturers of neonatal screening reagents and kits. The materials are valid until December 2029, and they replace the sixth ISNS-RPNS, which expired at the end of 2024. It is the hope of ISNS that manufacturers take advantage of this service and use these secondary reference materials to assign values to their DBS calibration materials for TSH, 17OHP, and PHE.

## 5. Technical Details

Contents: Each sachet contains a blood spot paper card with two sets of five calibrators (A–E) and desiccant.Storage: −20 °C with desiccant in Ziplock containers.Expiry date: 2029-12 (December 2029).Caution: The preparation contains material of human origin, which has been tested and found negative for HBsAg, HCV antibody, and HIV antibody. As with all materials of biological origin, this preparation should be regarded as potentially hazardous to health. It should be used and discarded according to your own laboratory’s safety procedures.Paper used for blood spot cards: The paper for the cards was supplied by Eastern Business Forms (Greenville, SC, USA).Product liability: ISNS accepts no liability whatsoever for any loss or damage arising from the use of this product, whether loss or profits, or indirect or consequential loss or otherwise, including, but not limited to, personal injury other than as caused by the negligence of ISNS.Citation: Please cite the title of the preparation (“7th ISNS-RPNS”) in all publications or data sheets for immunoassay kits in which this material is used for calibration.

## Figures and Tables

**Table 1 IJNS-11-00013-t001:** Initiation and expiry dates of the seven productions of the ISNS-RPNS.

	Prepared	Expiry
1st ISNS RPNS	2004	2007-12
2nd ISNS RPNS	2007	2010-12
3rd ISNS RPNS	2010	2013-12
4th ISNS RPNS	2013	2016-12
5th ISNS RPNS	2016	2019-12
6th ISNS RPNS	2019	2024-12
7th ISNS RPNS	2024	2029-12

**Table 2 IJNS-11-00013-t002:** Analyte enrichment for each calibrator in the 7th ISNS-RPNS.

Calibrator	A	B	C	D	E	Units
TSH	0	5	10	25	75	mIU/L blood
PHE	0	200	300	500	1000	µmol/L blood
17OHP	0	25	50	100	275	nmol/L blood

## Data Availability

For more information and additional analytical data, please can contact ISNS-Office: Dr. Peter Schielen, Manager, Reigerskamp 273, 3607HP Maarssen, The Netherlands, Email: peter.schielen@isns-neoscreening.org.

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
