# Peer review of "Seventh ISNS Reference Preparation for Neonatal Screening for Thyroid Stimulating Hormone, Phenylalanine, and 17α-Hydroxyprogesterone in Blood Spots"

_2409-515X, 2025, doi:10.3390/ijns11010013_

Round 1

Reviewer 1 Report

Comments and Suggestions for Authors

In this technical note “7th ISNS Reference Preparation for Neonatal Screening for TSH, Phenylalanine and 17α-Hydroxyprogesterone in Blood Spots” the authors describe the manufacture of the 7th combined dried blood spot reference preparation for neonatal screening (RPNS) for TSH, Phenylalanine and 17α-Hydroxyprogesterone. The note is well-written and provides a detailed description, with reference to the original publication, of the preparation and testing of the calibrator material for these analytes. Two minor comments:

  1. The reference numbering needs correction. There is only one reference, but it has been cited as reference number 2.
  2. For phenylalanine, the recovery is somewhat low (90.3–96.3%). While this is likely not clinically significant, was this low recovery observed in earlier material preparations? If it is significant, do the authors suggest reassigning the values in their Table or providing guidance for the user?

Author Response

Dear reviewer,

Thank you very much for taking the time to review this manuscript. Please find the detailed responses below and the corresponding revisions/corrections highlighted/in track changes in the re-submitted file.

Comment 1: The reference numbering needs correction. There is only one reference, but it has been cited as reference number 2.

Response 1: Thank you for pointing this out. We have corrected the reference numbers (and added one references as per suggestion of reviewer 2)

Comment 2: For phenylalanine, the recovery is somewhat low (90.3–96.3%). While this is likely not clinically significant, was this low recovery observed in earlier material preparations? If it is significant, do the authors suggest reassigning the values in their Table or providing guidance for the user?

Response 2: We have gone back to earlier reference materials (RPNS 1-6) and these recoveries are not uncommon. We feel that the values should not be reassigned but do feel that more information is needed for proper interpretation of results and the intention of the samples. We have now added text to that effect (line 63-70).

We hope that with this we have covered your comments sufficiently.

Sincerely Yours

Peter Schielen

Reviewer 2 Report

Comments and Suggestions for Authors

This paper is written well and succinctly describes the preparation of a combined dried blood spot Reference Preparation for Neonatal Screening that can assist manufacturers producing calibration materials for neonatal screening reagents.  The presentation is clear, and my only suggestion is that the authors include a reference in Section 3 Analytical Recovery that has more analytical information, such as De Jesus VR, Mei JV, Bell CJ, Hannon WH. Improving and assuring newborn screening laboratory quality worldwide: 30-year experience at the Centers for Disease Control and Prevention. Seminars in Perinatology. 34(2):125-33, 2010.  

Author Response

Dear reviewer,

Thank you very much for taking the time to review this manuscript. We have included your thoughtful suggestion to include the suggested reference-please see line 70 in the revised version of our document.

Sincerely Yours,

Peter Schielen